# Tracking the prevalence and emergence of SARS-CoV-2 variants of concern using a regional genomic surveillance program

Ana Jung,[1,2] Lindsay Droit,[1,2] Binita Febles,[1,3] Catarina Fronick,[4] Lisa Cook,[4] Scott A. Handley,[1,2] Bijal A. Parikh,[1] David Wang[1,3]

**ABSTRACT**  SARS-CoV-2 molecular testing coupled with whole-genome sequencing is instrumental for real-time genomic surveillance. Genomic surveillance is critical for monitoring the spread of variants of concern (VOCs) as well as discovery of novel variants. Since the beginning of the pandemic, millions of SARS-CoV-2 genomes have been deposited into public sequence databases. This is the result of efforts of both national and regional diagnostic laboratories. In this study, we describe the results of SARS-CoV-2 genomic surveillance from February 2021 to June 2022 at a metropolitan hospital in the United States. We demonstrate that consistent daily sampling is sufficient to track the regional prevalence and emergence of VOCs and recapitulate national trends. Similar sampling efforts should be considered a viable option for local SARS-CoV-2 genomic surveillance at other regional laboratories.

**IMPORTANCE** In our manuscript, we describe the results of SARS-CoV-2 genomic surveillance from February 2021 to June 2022 at a metropolitan hospital in the United States. We demonstrate that consistent daily sampling is sufficient to track the regional prevalence and emergence of variants of concern (VOCs). Similar sampling efforts should be considered a viable option for local SARS-CoV-2 genomic surveillance at other regional laboratories. While the SARS-CoV-2 pandemic has evolved into a more endemic form, we still believe that additional real-world information about sampling, procedures, and data interpretation is valuable for ongoing as well as future genomic surveillance efforts. Our study should be of substantial interest to clinical virologists.

**KEYWORDS**  SARS-CoV-2, ARTIC sequencing, genomic epidemiology

S ARS-CoV-2 first appeared in Wuhan, China, in late 2019 and was declared a global pandemic by the World Health Organization (WHO) on 11th of March, 2020 (1). The first SARS-CoV-2 genome sequence was determined in January of 2020 (2). Since then, over 15 million SARS-CoV-2 genomes have been sequenced and made publicly available (3, 4). This global genomic surveillance project has been an effective way to identify and track nucleotide changes with the potential to influence viral transmission dynamics, pathogenicity, diagnostic performance, vaccine efficacy, and immune escape (2, 5, 6).

Genomic surveillance has enabled classification of emergent SARS-CoV-2 into variants of concern (VOCs), variants of interest (VOIs), variants being monitored (VBMs), and variants of high consequence (VOHCs). This classification is based on their predicted transmissibility, virulence, and ability to cause severe disease. Classification of SARS-CoV-2 variants is changing over time. Previously classified SARS-CoV-2 VOCs include Alpha (B.1.1.7), Beta (B.1.351), Gamma (P.1), Delta (B.1.617.2), and Omicron (B.1.1.529); VOIs include Lambda (C.37) and Mu (B.1.621); VBMs include AZ.5, C.1.2, B.1.617.1*, B.1.526*, B.1.525*, B.1.630, and B.1.640 (7). As of December 1, 2022, the only VOC lineage is Omicron, with Omicron XBB.1.5 being the only VOI as of March 15, 2023.

Address correspondence to David Wang, davewang@wustl.edu.

The authors declare no conflict of interest.

See the funding table on p. 10.

Information about which VOCs are circulating within a regional population is important for public health preparedness and response. In addition, regional genomic surveillance can lead to the original identification of many important VOCs. This includes the original detection of Alpha, Beta, Gamma, Delta, and Omicron, which were first detected in the United Kingdom, South Africa, Brazil, India, and multiple countries, respectively (8). Implementing a regional genomic surveillance program requires significant expense, time, and access to modern sequencing technology and bioinformatics expertise. Thoughtful sample selection (size and frequency) is critical for creating a regional genomic surveillance program capable of detecting circulating VOCs and for discovery of novel variants within the confines of local resources.

In this study, we report the results of a regional SARS-CoV-2 genomic surveillance program run at a metropolitan hospital in St. Louis, MO, USA, at a sampling of ~5 samples/1,000,000 people/week ($n$ = 1,240). Our findings provide evidence that, using this sampling rate, we were able to generate regional variant profiles similar to national trends. These results serve as an example on how moderate SARS-CoV-2 genomics surveillance programs can provide useful regional information. This is particularly important for setting up surveillance programs in low-resource settings.

## MATERIALS AND METHODS

### Sample collection

Nasopharygeal (NP) swab specimens were collected in universal transport medium and submitted for routine clinical SARS-CoV-2 testing at the Barnes-Jewish Hospital Molecular Infectious Disease Lab. Barnes-Jewish Hospital is a 1,400-bed nonprofit teaching hospital—the largest in Missouri. It services the St. Louis metropolitan area (population 2.8 million, the 21[st] largest city in the United States as of 2020). From 1 February 2021 through 30 June 2022, 27,886 positive tests were recorded through various testing methods in the clinical labs. Sequencing selection was based on samples with sufficient viral load, as determined by testing on the Roche cobas 6800 instruments (Roche) according to the manufacturer's instructions. These instruments performed the majority of testing and demonstrated 18,125 positive results during this time period. The number of sequenced variants, 1,240, represents 6.8% of positive results from the instruments the samples were collected on and 4.4% of all positive tests handled in the clinical laboratory. Samples positive for SARS-CoV-2 with a minimum cycle threshold (Ct) of 27 for either of two assay targets were eligible for genomic sequencing. A total of three to four random specimens per day, from the pool of all eligible specimens with a sufficient residual volume, were ultimately selected for archival and subsequent sequence analysis. Samples were archived at −70°C in cryovials between 3 and 7 days post-collection.

### SARS-CoV-2 genome sequencing

Total nucleic acid was extracted on a MagNa Pure instrument (Roche) according to the manufacturer's recommendations. cDNA was prepared using the ARTIC v3 protocol for samples collected between the dates of February and October 2021 and the ARTIC v4 protocol between the dates of November 2021 and June 2022 (9, 10). The cDNA was purified by a 1 x AMPure bead cleanup with a final elution in 10 mM TrisHCl, pH 8.5. Purified cDNA was quantitated by a Qubit 1 x dsDNA HS Assay (Thermofisher). Fifty to one hundred nanograms of the 400-bp cDNA amplicons were converted into Illumina libraries on the Ep5075 (Eppendorf) using the KAPA HyperPrep kit (Roche Diagnostics) using one-fourth of vendor-recommended reagents and full-length dual-indexed adapters diluted to 250 nM (11). Final libraries were checked for quality and quantity on the LabChipGX instrument (PerkinElmer) using the DNA High-sensitivity kit. Libraries were normalized to 5 nM, and an equal volume was pooled per library. This final library pool was quantitated by qPCR using the KAPA Library Quantification kit (Roche

Diagnostics) and diluted to 2 nM for sequencing in 10 mM TrisHCl, pH 8.5. Libraries were loaded at 12pM with a 20% PhiX spike in on the MiSeq v3, 600 cycle kit according to Illumina's guidelines, generating 2 × 250 reads.

## Analysis of SARS-CoV-2 genomic sequences

We implemented a SARS-CoV-2 genome analysis pipeline that started with raw sequence data and generated quality control information, consensus genomes using the Chan Zuckerberg Biohub (CZ Biohub) SARS-CoV-2 Illumina Pipeline (https://github.com/czbiohub/sc2-illumina-pipeline). Polymorphisms, insertions, and deletions were determined using the default settings in Minimap2 as implemented in the CZ Biohub pipeline. Consensus genome lineage assignments were created using both Nextclade (v.2.9.1) and Pangolin (v.4.1.3) (12, 13). Per run phylogenetic trees were generated using Augur and visualized in Microreact (14, 15).

## RESULTS

### SARS-CoV-2 genome assessment

Illumina sequences obtained using the ARTIC protocol were processed through our customized SARS-CoV-2 genome analysis pipeline (Fig. 1). This workflow generates consensus genomes and lineage assignments using both Pangolin and Nextclade, respectively. Missing data are assessed using customized plots (Fig. 1B). Missing data plots were used to assess the genomic location of missing data (basecall = N) due to either poor sequence quality or primer dropout. These plots are useful for assessing if a sample failed to amplify (extensive coloring across the plot) or experienced single primer (repeated pattern) or localized (short stretches across samples) dropouts. Consensus genomes were excluded from downstream analysis if they were more than 1,000 bases shorter than the 29,903-bp SARS-CoV-2 Wuhan-Hu-1 reference genome. Interactive phylogenetic trees (Fig. 1C) are also created and visualized on Microreact (https://

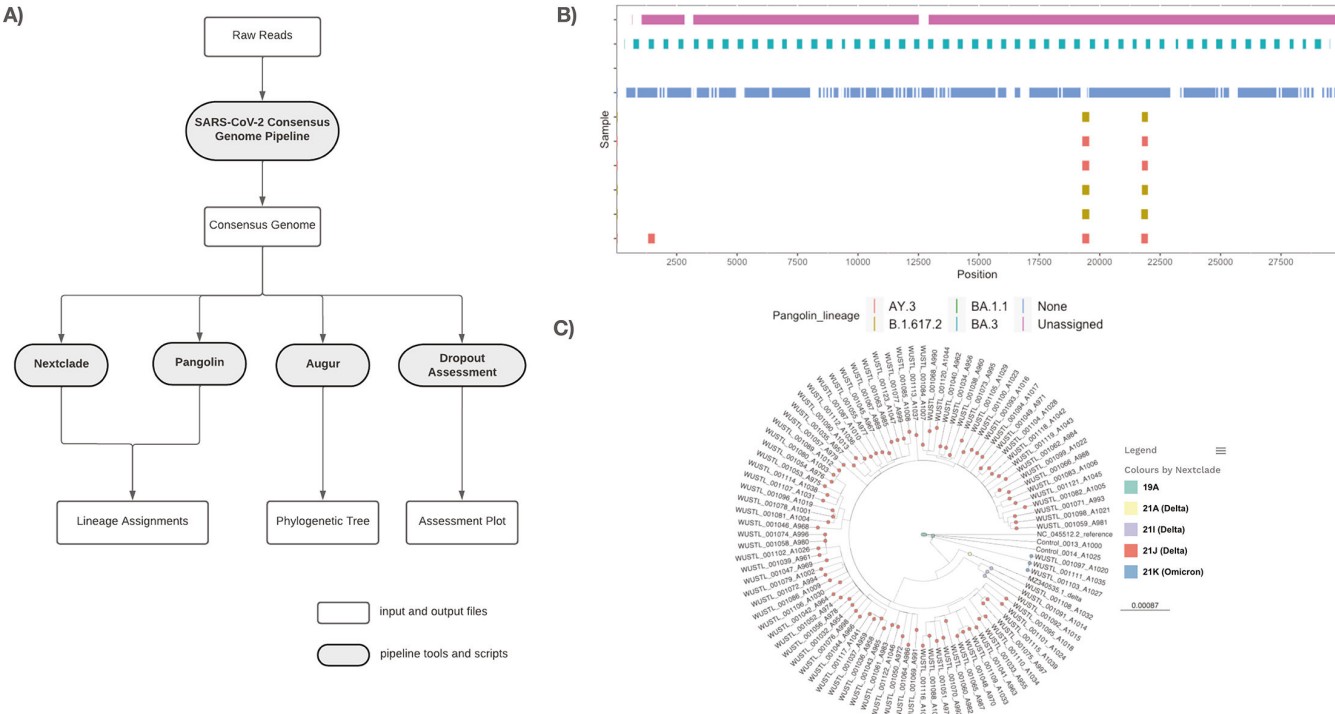

**FIG 1** (A) SARS-CoV-2 genome analysis pipeline. (B) "Dropout Assessment" location of missing data (Ns) within a selected subset of SARS-CoV-2 consensus genomes. Colors indicate missing data in specific Pangolin lineage assignments. (C) Single run phylogenetic tree of SARS-CoV-2 consensus genomes visualized using Microreact.

microreact.org/). In total, we performed ARTIC sequencing on 1,540 samples, from which 1,240 consensus genomes passed the size threshold criteria. These consensus genomes were included in all subsequent analyses.

## Regional versus national VOC prevalence

Consensus genome sequences were classified based on their similarity to known VOCs. The number of VOCs identified biweekly were calculated and compared to national proportions, as reported by the Centers for Disease Control and Prevention (CDC) (SARS-CoV-2 Variant Proportions) (Fig. 2). Regional VOC prevalence exhibited a great deal of parity with national proportions. During the earliest phases of the pandemic (winter through mid-summer of 2021), genomes with similarity to the Wuhan-Hu-1 reference genome were gradually replaced with strains from the Alpha lineage. During this same period, the Beta lineage of SARS-CoV-2 subtly emerged nationally, but it was not seen in our local genomic surveillance. The Delta lineage was first observed nationally during the first 2 weeks of May 2021, but regional detection was slightly delayed until the last 2 weeks. Other than detection of a limited amount of the Alpha lineage, the Delta lineage had completely taken over both regionally and nationally by mid-July 2021. Omicron was detected both regionally and nationally in early December and completely replaced the Delta variant by the end of January, 2022. Omicron remained the dominant VOC throughout the spring and summer of 2022.

## Monitoring regional SARS-CoV-2 subvariant lineages

SARS-CoV-2 VOC lineages are composed of a collection of subvariant lineages. In particular, early pandemic spread of the Omicron lineage was characterized by two primary subvariant lineages. Omicron subvariant lineage BA.1 dominated 2021, with subvariant BA.2 emerging during the winter of 2021/2022 (7). Our regional genomic

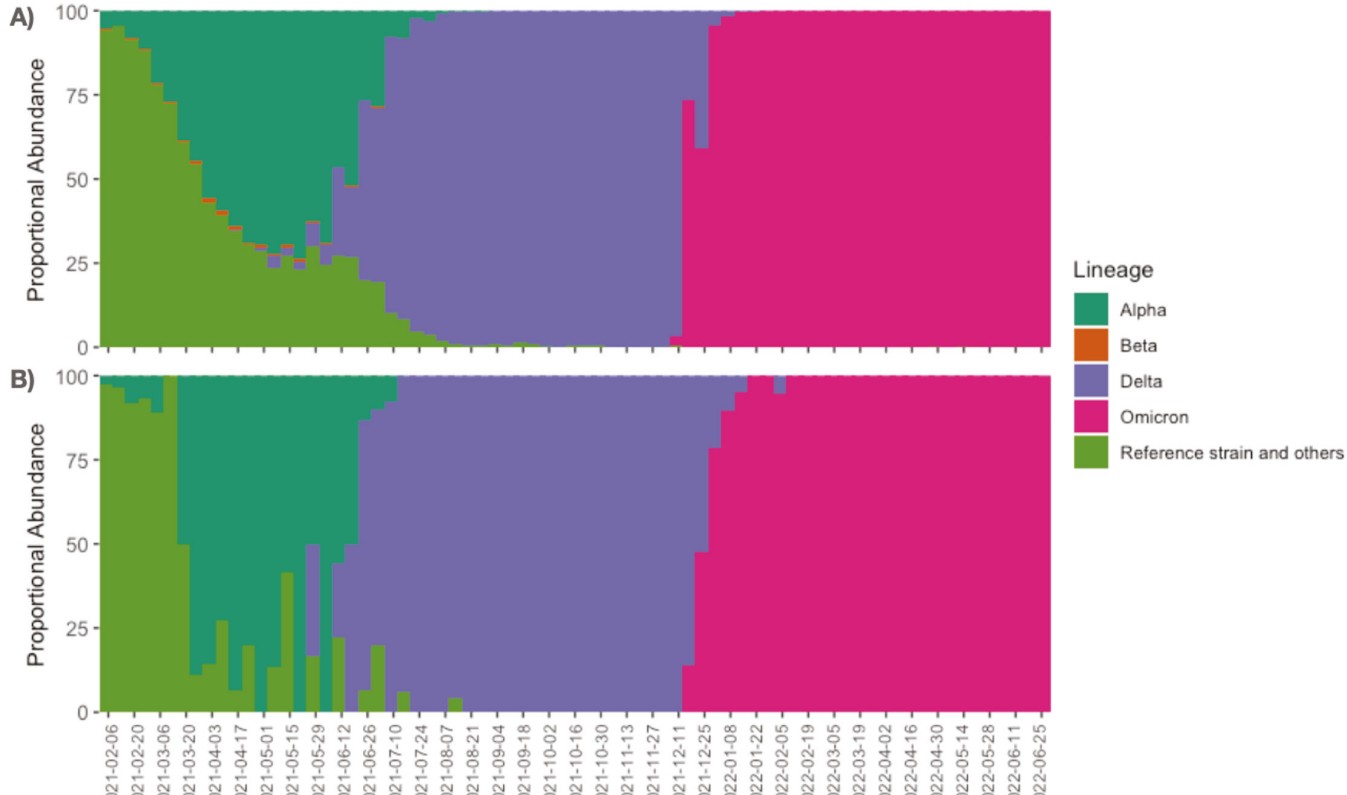

**FIG 2**  (A) National VOC prevalence as reported by the Centers for Disease Control and Prevention (CDC). (B) Regional VOC prevalence as detected in the current study.

surveillance identified similar patterns (Fig. 3). The only local Omicron subvariants identified between December 2021 and February 2022 belonged to the BA.1 lineage (BA1, BA1.1, BA.1.1.10, BA.1.4, and BA.1.15). Omicron subvariant BA.2 was first detected in February 2022, completely replacing the BA.1 subvariant lineage by the end of April 2022. BA.2 was dominant throughout the summer of 2022 with the dominant subvariants belonging to BA.2 and BA.2.12.1. In total, 17 BA.2 subvariants were detected throughout this time period. In addition, Omicron subvariant lineages BA.2 and BA.5 began to be detected in April/May 2022, with increasing detection of BA.5.5 through June.

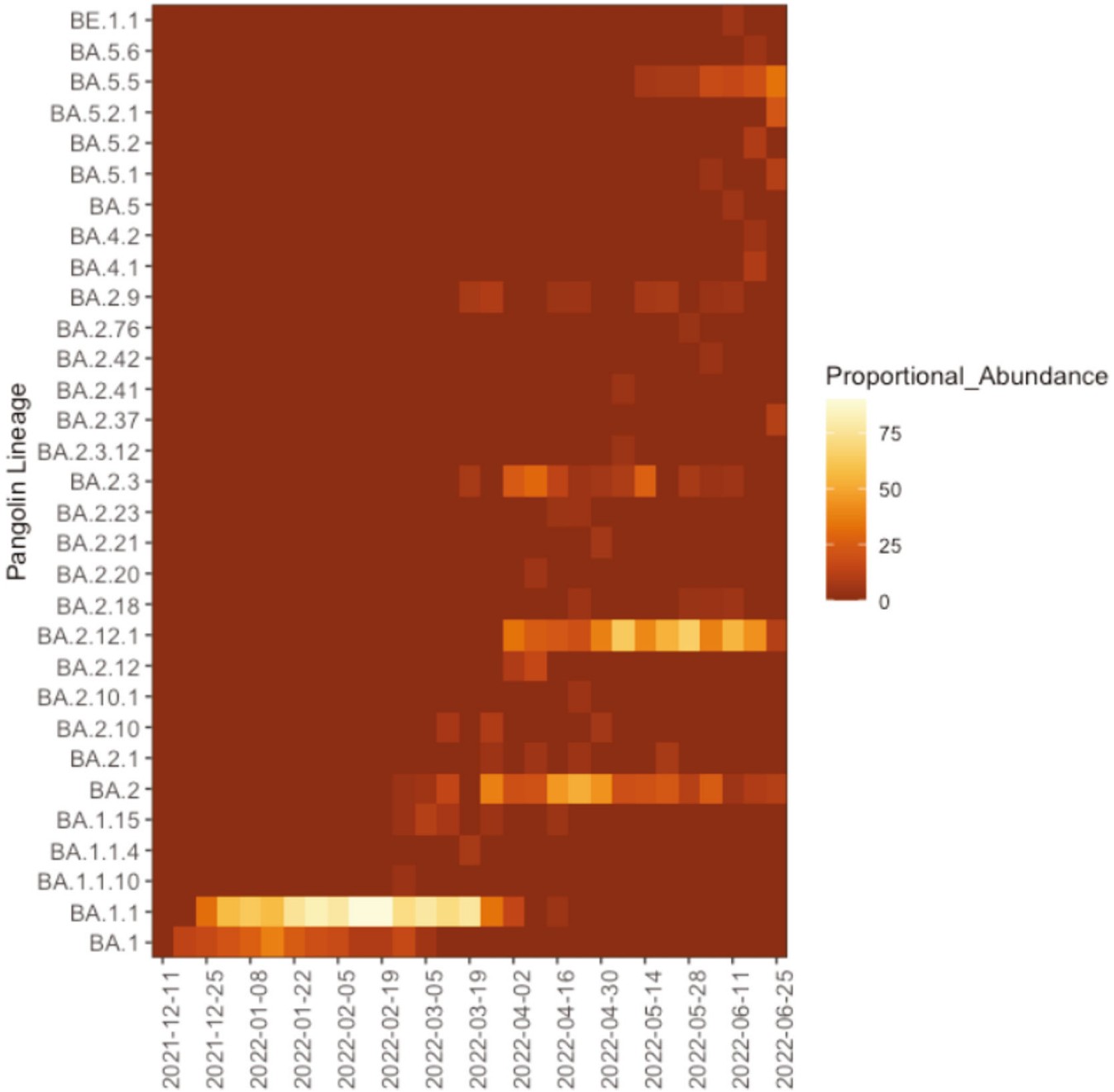

**FIG 3** Detection of local SARS-CoV-2 Omicron subvariant lineages between December 2021 and June 2022.

## Analysis of regional genomic sequence divergence from known global references

Sequence divergence from all locally acquired consensus genomes and selected reference genomes was calculated using Nextclade (Fig. 4) (13). All variants were calculated descendants of previously known reference variants. All local consensus genomes clustering within the Alpha or reference-like clades were classified as descendants to B.1.2, B.1.1.7, BA.1.1.519, and several other B.1.X and B.1.1.X sublineages (Fig. 4A). All local consensus genomes assigned to the Delta and Omicron VOC were assigned to the 21A or 21J (Fig. 4B) and BA.1, BA.2, BA.4, and BA.5 (Fig. 4C) subvariant lineages. No local consensus genomes clustered with WHO reference recombinant genomes.

## Analysis of deletions and insertions in regional SARS-CoV-2 genomes

We assessed the prevalence and length of deletions and insertions in all local consensus genomes, relative to the SARS-CoV-2 Wuhan-Hu-1 reference genome. We identified a large number of genomic deletions of various frequencies and lengths (Fig. 5; Table S2). High-frequency deletions (> 75% frequency) were identified in all lineages of local consensus genomes other than those assigned to the reference strain (Wuhan-Hu-1) lineage where the deletion frequency was relatively low (Fig. 5A). The number of deletions per lineage varied with genomes assigned to the reference strain ($n = 17$), Alpha ($n = 8$), Delta ($n = 47$), and Omicron lineages (21K = 15 and 21L = 19) (Fig. 5). High-frequency (> 75% frequency) deletions occurred most frequently in ORF1ab and the S gene in genomes assigned to the Alpha, Delta, and Omicron lineages. Lineage-specific high-frequency deletions were identified in the N gene of Alpha and Omicron (21K and 21L), ORF8 of Delta, and the 3' UTR of Omicron 21L. The minimum deletion length

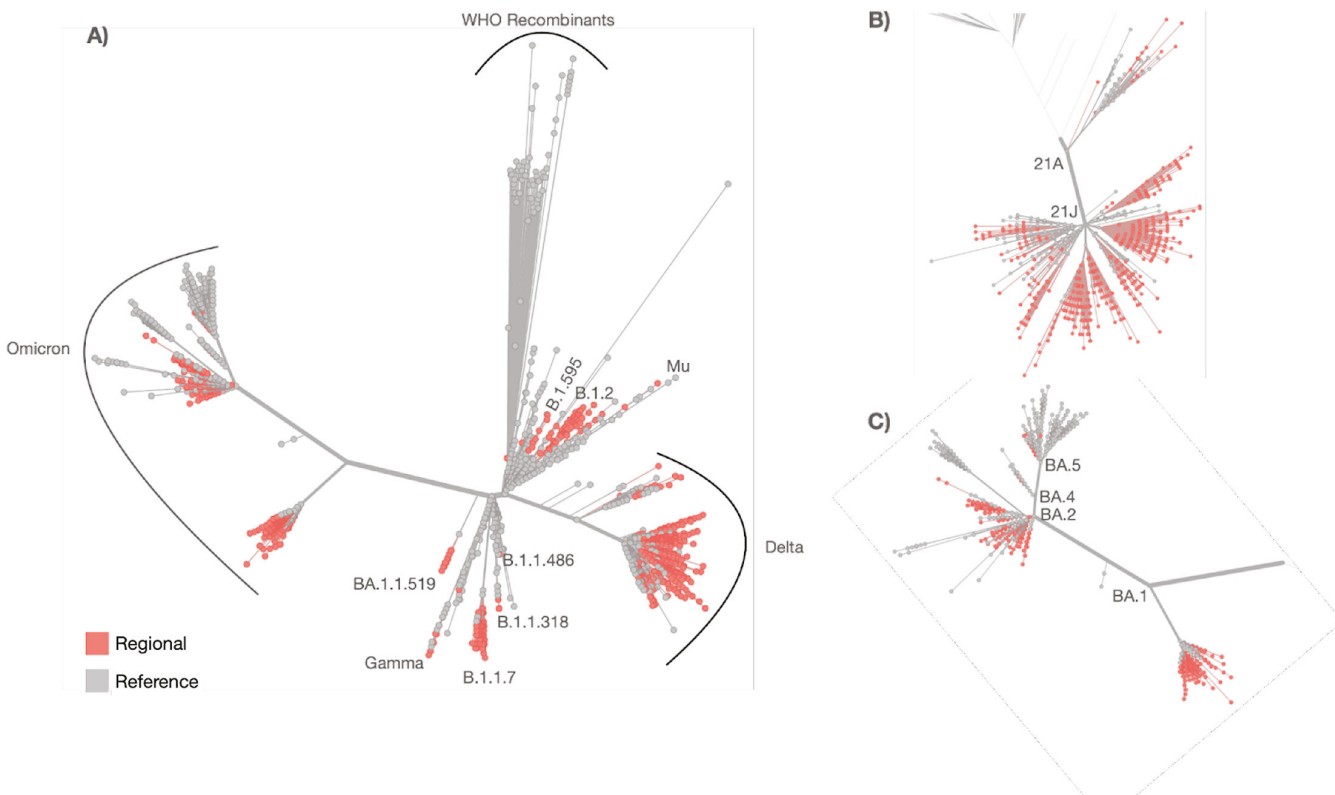

**FIG 4**  (A) Phylogenetic tree representing regional sequence divergence in relation to global reference VOC genomes. Zoomed-in representation of local sequence variants in relation to global reference genomes for (B) Delta and (C) Omicron lineages.

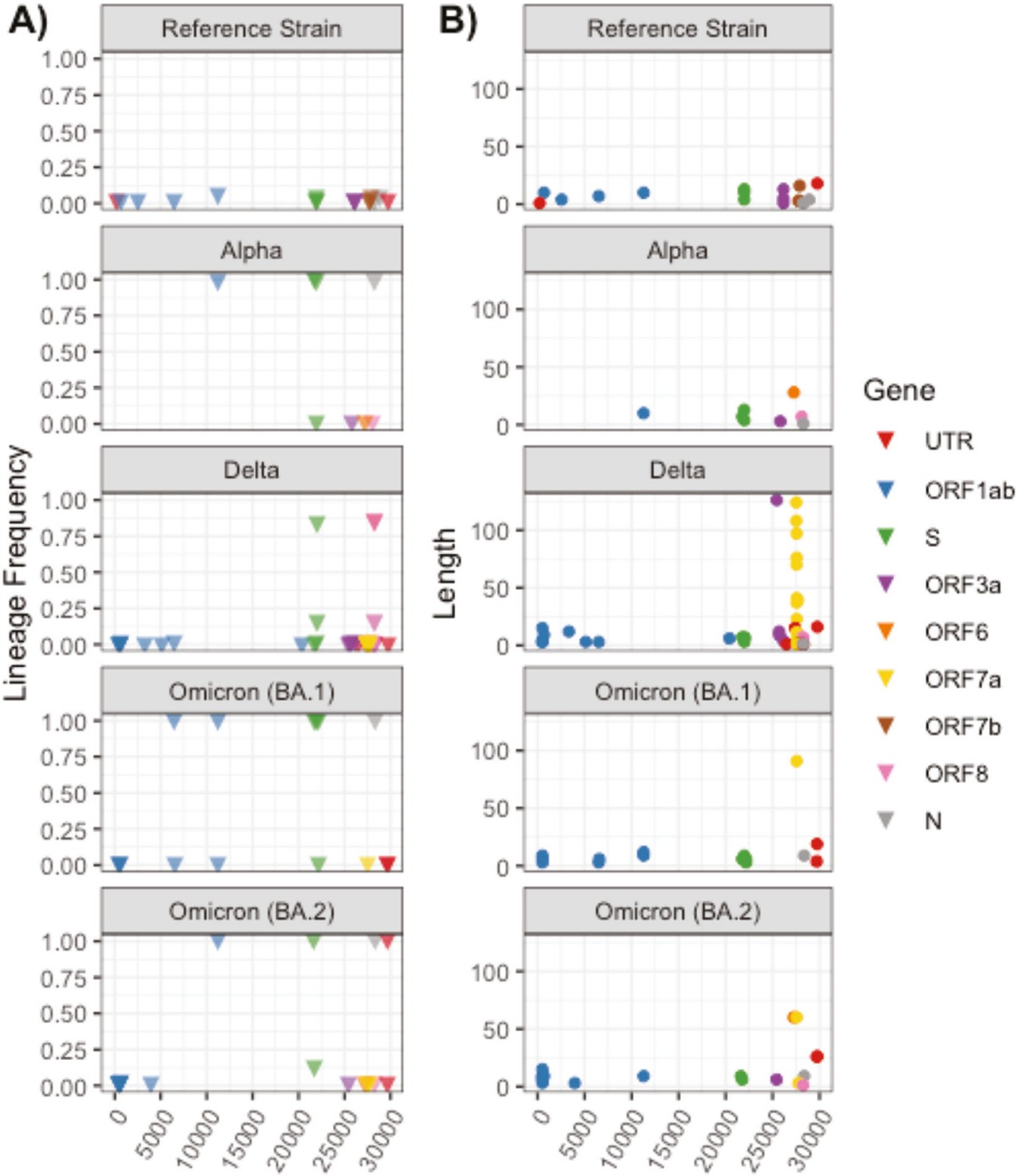

**FIG 5** (A) Lineage-specific deletion frequency and (B) deletion length in local consensus genomes.

across all consensus genomes was 1 base with a maximum deletion length of 126 bases in the Delta lineage within Orf3a (Fig. 5B). We identified a cluster of deletions within Orf7a of the Delta lineage (Fig. 5B). These Delta lineage-specific Orf7a deletions had an average start at position 27,588 (min = 27,520; max = 27,721) and an average end at

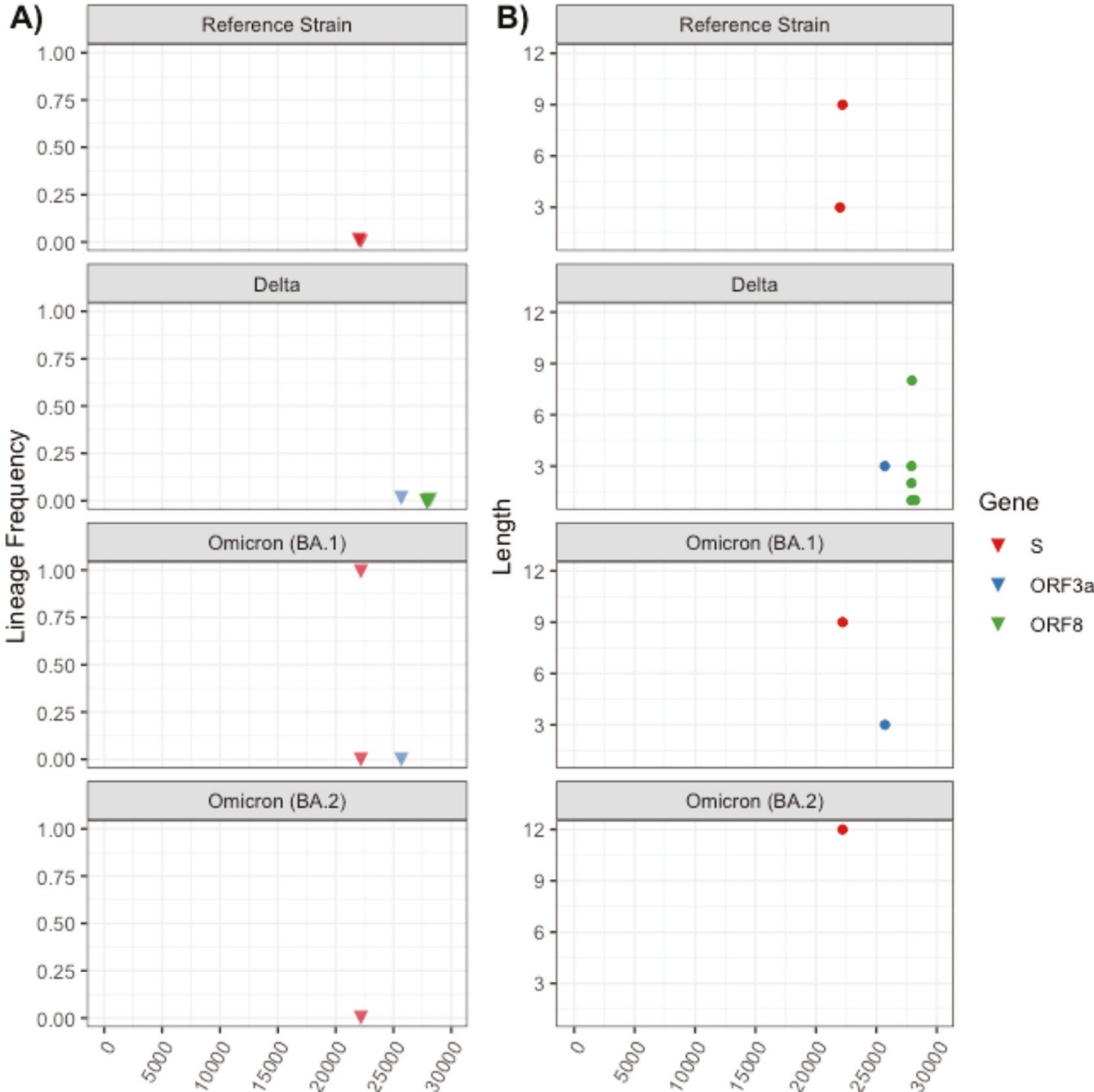

FIG 6   (A) Lineage-specific insertion frequency and (B) length in local consensus genomes.

position 27,631 (min = 27,577; max = 27,741), yielding deletions ranging from 1 to 124 bases in length with an average length of 43.5 bases. Each Orf7a deletion was only found in one or two out of 490 local consensus SARS-CoV-2 Delta genomes (lineage frequency range = 0.002%–0.004%), with the largest 124-base deletion only detected once.

Relative to deletions, far fewer insertions were identified in local consensus genomes (Fig. 6; Table S2). There were no insertions found in consensus genomes assigned to the Alpha lineage. Insertions in the S gene were found in consensus genomes assigned to the reference strain and Omicron strains but were absent for genomes assigned to the Delta lineage. Insertions ranged in length from 1 to 12 bases, with the longest insertion occurring in the S gene of a single Omicron (BA.2) consensus genome.

## DISCUSSION

Pandemics are a global phenomenon comprising a network of local, community-driven outbreaks. Understanding global transmission starts at the local level. Conversely, understanding how local infection dynamics can be compared to global trends can provide regional information about the emergence and circulation of globally identified strains. In addition, genomic surveillance can provide highly detailed genetic maps of regionally circulating strains.

Building a local genomic surveillance program requires access to the appropriate infrastructure including a patient-facing diagnostic laboratory for detection and procurement of samples containing the pathogen of interest, a molecular biology laboratory, modern sequencing infrastructure, and bioinformaticians for data processing and analysis. Availability to these resources will vary from region-to-region, but a universal consideration will be sampling depth and cadence. Ultimately, sampling will depend on the goals of the genomic surveillance program as well as financial considerations and laboratory throughput. While access to resources and financing varies across regions, it is still useful to place some boundaries for building a successful sampling framework.

In the present study, we assessed our ability to detect globally circulating SARS-CoV-2 VOC in a metropolitan hospital in St. Louis, MO, USA. We sampled ~5 samples/1,000,000 people per week ($n = 1,240$ samples), half of the requirement that a predicted model suggests (16). We detected all significant SARS-CoV-2 VOCs (Alpha, Delta, and Omicron) and were also able to detect sublineage variations circulating within the community. Our data support that low number consistent sampling is sufficient to detect the prevalence of known and novel VOCs within a community.

In addition to determining local VOC prevalence, regional genomic surveillance provides a wealth of detailed genetic information. While low-frequency subvariants may only appear transiently in a region and never achieve global transmission, they still contain genomic variation useful for mechanistic studies. For example, a 115-base pair deletion in SARS-CoV-2 ORF7a (27,549–27,644 nt) was identified as part of regional genomic sequencing efforts in Bozeman, Montana, USA (17). This ORF7a deletion (ORF7a$^{\Delta115}$) was subsequently shown to have an *in vitro* growth defect associated with an elevated IFN response, suggesting an immunosuppressive role for ORF7a. In our study, we also identified a collection of deletions in the Orf7a gene in strains from the Delta lineage. Deletions in Orf7a were uncommon, but were as large as 124 bases long. These Orf7a and other subvariant isolates can serve as a natural laboratory bridging real-world genetic variation with detailed mechanistic studies.

Taken together, we demonstrate that modest sampling efforts paired with robust infrastructure can provide genomic surveillance of SARS-CoV-2 at a regional level. These efforts can characterize the presence and prevalence of VOCs. We believe these findings are particularly important when considering setting up surveillance programs in low-resource settings. We demonstrate that with a limited sampling of sequences, we were able to recapitulate sequencing efforts performed at a national level. While this may not prove true for the transmission of all diseases, it does indicate that small, regional efforts are worthwhile and may be more achievable in low-resource settings. Non-genomics methods, such as RT-PCR, which can readily detect the presence of SARS-CoV-2, should also be considered during the study design process. This method lacks the ability to resolve base-level sequence variation across the genome. Still, these techniques are incredibly efficient in detecting the presence of a virus at considerable cost savings and efficiency. In addition, regional genomic surveillance generates a repository of unique genetic variations. We believe these results should serve as a guideline for future SARS-CoV-2 genomic surveillance programs.

## ACKNOWLEDGMENTS

This study was funded by the NIH grant U01 AI151810.

B.A.P. and D.W. conceptualized the study. B.A.P. coordinated all sample collections. L.D., C.F., and L.C. organized sample extraction and sequencing. A.J., B.F., and S.A.H. completed all bioinformatics analyses and figure design and creation. Writing was shared evenly across all authors.

## AUTHOR AFFILIATIONS

[1]Department of Pathology & Immunology, Washington University School of Medicine, St. Louis, Missouri, USA

[2]The Edison Family Center for Genome Sciences & Systems Biology, Washington University School of Medicine, St. Louis, Missouri, USA

[3]Department of Molecular Microbiology, Washington University School of Medicine, St. Louis, Missouri, USA

[4]McDonnell Genome Institute, Washington University School of Medicine, St. Louis, Missouri, USA

## AUTHOR ORCIDs

Scott A. Handley  http://orcid.org/0000-0002-2143-6570
Bijal A. Parikh  http://orcid.org/0000-0003-2490-8294
David Wang  http://orcid.org/0000-0002-0827-196X

## FUNDING

| Funder | Grant(s) | Author(s) |
| --- | --- | --- |
| HHS | National Institutes of Health (NIH) | AI151810 | Scott A. Handley |
| | | David Wang |

## AUTHOR CONTRIBUTIONS

Ana Jung, Formal analysis, Investigation, Methodology, Writing – original draft, Writing – review and editing | Lindsay Droit, Formal analysis, Investigation, Methodology, Writing – review and editing | Binita Febles, Data curation, Formal analysis, Investigation, Visualization, Writing – review and editing | Catarina Fronick, Formal analysis, Investigation, Methodology, Writing – review and editing | Lisa Cook, Investigation, Methodology, Writing – review and editing | Scott A. Handley, Data curation, Formal analysis, Investigation, Methodology, Project administration, Supervision, Validation, Visualization, Writing – original draft, Writing – review and editing | Bijal A. Parikh, Conceptualization, Formal analysis, Funding acquisition, Investigation, Methodology, Project administration, Resources, Supervision, Validation, Visualization, Writing – original draft, Writing – review and editing | David Wang, Conceptualization, Funding acquisition, Investigation, Methodology, Project administration, Supervision, Validation, Visualization, Writing – original draft, Writing – review and editing

## DATA AVAILABILITY

Whole viral genomes were made publicly available at GenBank (Table S1), and raw data are available at NCBI BioProject PRJNA748401.

## ETHICS APPROVAL

Ethical approval for this study was obtained from the Washington University School of Medicine Institutional Review Board (IRB202004259) with a waiver of consent. Specimens were accessed beginning 30 April 2020 through 1 July 2022. Authors had temporary access to information that could identify individual participants, as approved by the IRB.

## ADDITIONAL FILES

The following material is available online.

### Supplemental Material

**Table S1 (Spectrum04225-23-s0001.xlsx).** Genomes deposited in GenBank.
**Table S2 (Spectrum04225-23-s0002.xlsx).** List of indels observed.

### Open Peer Review

**PEER REVIEW HISTORY (review-history.pdf).** An accounting of the reviewer comments and feedback.

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
