## [Reviewer comments · Microbiology Spectrum]

Microbiology Spectrum

Tracking the prevalence and emergence of SARS CoV2 variants of concern using a regional genomic surveillance program

Ana Jung, Lindsay Droit, Binita Febles, Catrina Fronick, Lisa Cook, Scott Handley, Bijal Parikh, and David Wang

Corresponding Author(s): David Wang, Washington University in St Louis School of Medicine

Review Timeline:

Submission Date:	December 19, 2023
Editorial Decision:	January 31, 2024
Revision Received:	April 17, 2024
Accepted:	May 14, 2024

Editor: Holly Ramage

Reviewer(s): The reviewers have opted to remain anonymous.

Transaction Report:

DOI: <https://doi.org/10.1128/spectrum.04225-23>

Re: Spectrum04225-23 (Tracking the prevalence and emergence of SARS CoV2 variants of concern using a regional genomic surveillance program)

Dear Dr. David Wang:

Thank you for the privilege of reviewing your work. Below you will find my comments, instructions from the Spectrum editorial office, and the reviewer comments.

Your manuscript was evaluated by two expert referees. As you will read, both reviewers appreciated the significance of your work and the contribution of your study to SARS-CoV-2 genomic surveillance efforts. The reviewers have provided several thoughtful comments and suggestions to improve the manuscript and incorporate your findings in the context of broader surveillance efforts. In addition, a clear justification regarding patient sampling and whether this strategy is sufficient to detect prevalence of variants of concern in a regional context would better support your conclusions.

Revision Guidelines

Sincerely,
Holly Ramage
Editor
Microbiology Spectrum

Reviewer #1 (Comments for the Author):

Thank you for the opportunity to review this interesting study. The manuscript is well-written, and contributes valuable insights to SARS-CoV-2 genomic surveillance. The authors have presented a comprehensive study on SARS-CoV-2 genomic surveillance at a metropolitan hospital in the USA from February 2021 to June 2022. Their findings highlight the significance of consistent daily sampling in tracking regional prevalence and the emergence of variants of concern (VOC). However, the discussion section needs some improvements.

Strengths:

- The study design is comprehensive, covering a substantial timeframe and utilizing robust methods for sample collection, sequencing, and analysis.
- The methods section is well-detailed, providing clarity on ethical considerations, sample collection, sequencing protocols, and data analysis pipelines.
- The authors conducted a detailed analysis of the SARS-CoV-2 genomic sequences, including classification of VOCs, comparison with national prevalence, identification of sub_variant lineages, and analysis of genetic variations.
- The manuscript emphasizes the importance of regional genomic surveillance for public health preparedness and response. The findings contribute to understanding the prevalence and dynamics of SARS-CoV-2 variants at the local level.

Areas for Improvement:

1. The manuscript could benefit from discussion on the challenges and potential strategies for implementing genomic surveillance in low-resource settings. This would add a broader perspective to the study's implications.
2. While the manuscript focuses on whole genome sequencing, a brief discussion on the complementary role of RT-PCR in surveillance would enhance the completeness of the discussion section.
3. The manuscript could provide more specific recommendations for future SARS-CoV-2 genomic surveillance programs, offering guidance on optimal sampling strategies and adaptation to local resource capacities.

Few comments are provided as notes in the attached PDF file.

Reviewer #2 (Comments for the Author):

I think the authors could use the data in another way to provide more significant analytical support for their claims.

Tracking the prevalence and emergence of SARS CoV2 variants of concern using a regional genomic surveillance program

Ana Jung^{1,2}, Lindsay Droit^{1,2}, Binita Febles^{1,4}, Catarina Fronick³, Lisa Cook³, Scott A. Handley^{1,2}, Bijal A Parikh¹ and David Wang^{1,4#}

1) Department of Pathology & Immunology, Washington University School of Medicine, St. Louis, MO, USA

2) The Edison Family Center for Genome Sciences & Systems Biology, Washington University School of Medicine, St. Louis, MO, USA

3) McDonnell Genome Institute, Washington University School of Medicine, St. Louis, MO, USA

4) Department of Molecular Microbiology, Washington University School of Medicine, St. Louis, Missouri, USA

Address correspondence to: David Wang (davewang@wustl.edu) Washington University

School of Medicine, 660 South Euclid Avenue, Campus Box 8118, St. Louis, MO 63110-1093

USA +1 (314) 273-7926.

Keyword: SARS-CoV-2; ARTIC sequencing, genomic epidemiology

ABSTRACT

SARS-CoV-2 molecular testing coupled with whole genome sequencing is instrumental for real-time genomic surveillance. Genomic surveillance is critical for monitoring the spread of variants of concern (VOC) as well as novel variant discovery. Since the beginning of the pandemic millions of SARS-CoV-2 genomes have been deposited into public sequence databases. This is the result of efforts of both national and regional diagnostic laboratories. Here we describe the results of SARS-CoV-2 genomic surveillance from February 2021 to June 2022 at a metropolitan hospital in the USA. We demonstrate that consistent daily sampling is sufficient to track the regional prevalence and emergence of VOC. Similar sampling efforts should be considered a viable option for local SARS-CoV-2 genomic surveillance at other regional laboratories.

INTRODUCTION

SARS-CoV-2 first appeared in Wuhan, China in late 2019 and was declared a global pandemic by the World Health Organization (WHO) on 11th of March, 2020 [1]. The first SARS-CoV-2 genome sequence was determined in January of 2020 [2]. Since this time over 15 million SARS-CoV-2 genomes have been sequenced and made publicly available [3][4]. This global genomic surveillance project has been an effective way to identify and track nucleotide changes with the potential to influence viral transmission dynamics, pathogenicity, diagnostic performance, vaccine efficacy and immune escape [2,5,6].

Genomic surveillance has enabled classification of emergent SARS-CoV-2 into variants of concern (VOC), variants of interest (VOI), variants being monitoring (VBM), and variants of high consequence (VOHC). This classification is based on their predicted transmissibility, virulence, and ability to cause severe disease. Classification of SARS-CoV-2 variants is changing overtime. Previously classified SARS-CoV-2 VOCs include Alpha (B.1.1.7), Beta (B.1.351), Gamma (P.1), Delta (B.1.617.2), and Omicron (B.1.1.529); VOI included Lambda (C.37) and Mu (B.1.621); and

VBM include AZ.5, C.1.2, B.1.617.1*, B.1.526*, B.1.525*, and B.1.630, B.1.640 [7]. As of December 1, 2022, the only VOC lineage is Omicron with Omicron XBB.1.5 being the only VOI as of March 15, 2023.

Information about which VOC are circulating within a regional population is important for public health preparedness and response. In addition, regional genomic surveillance can lead to the original identification of many important VOC. This includes the original detection of Alpha, Beta, Gamma, Delta and Omicron which were first detected in the United Kingdom, South Africa, Brazil, India and multiple countries, respectively [8]. Implementing a regional genomic surveillance program requires significant expense, time and access to modern sequencing technology and bioinformatic expertise. Thoughtful sample selection (size and frequency) is critical to creating a regional genomic surveillance program capable of detecting circulating VOC and novel variant discovery within the confines of local resources.

Here we report the results of a regional SARS-CoV-2 genomic surveillance program run at a metropolitan hospital in St. Louis, MO, USA at a sampling of ~5 samples/1,000,000 people/week (n = 1,240). Our findings provide evidence that this sampling rate is sufficient to detect the prevalence of known and novel VOC within a community. These results should serve as a guideline for future SARS-CoV-2 genomic surveillance programs.

MATERIALS AND METHODS

Ethical considerations. Ethical approval for this study was obtained from the Washington University School of Medicine Institutional Review Board (IRB202004259) with a waiver of consent. Specimens were accessed beginning 4/30/2020 through 7/01/2022. Authors had temporary access to information that could identify individual participants, as approved by the IRB.

Sample collection. Nasopharyngeal (NP) swab specimens were collected in universal transport medium and submitted for routine clinical SARS-CoV-2 testing at the Barnes-Jewish Hospital Molecular Infectious Disease Lab. Specimens were tested for the presence of SARS-CoV-2 on the cobas SARS-CoV-2 or cobas SARS-CoV-2 & influenza A/B assays performed on the cobas 6800 instrument (Roche) according to manufacturer instructions. Samples positive for SARS-CoV-2 with a minimum cycle threshold (Ct) of 27 for either of two assay targets were eligible for genomic sequencing. 3 to 4 random specimens per day, from the pool of all eligible specimens with sufficient residual volume, were ultimately selected for archival and subsequent sequence analysis. Samples were archived at -70°C in cryovials between 3 and 7 days post-collection.

SARS-CoV-2 genome sequencing. Total nucleic acid was extracted on a MagNa Pure instrument (Roche) according to the manufacturer recommendations. cDNA was prepared using the ARTIC v3 protocol for samples collected between the dates of February and October, 2021 and the ARTIC v4 protocol between the dates of November, 2021 and June, 2022 [9,10]. The cDNA was purified by a 1x AMPure bead cleanup with a final elution in 10mM TrisHCl, pH 8.5. Purified cDNA was quantitated by a Qubit 1x dsDNA HS Assay (Thermofisher). 50-100ng of the 400bp cDNA amplicons are converted into Illumina libraries on the Ep5075 (Eppendorf) using the KAPA Hyper library prep kit (Roche Diagnostics) using 1/4 of vendor recommended reagents and full length dual indexed adaptors diluted to 250nM [11]. Final libraries were checked for quality

and quantity on the LabChipGX instrument (Perkin Elmer), using the DNA High-sensitivity kit. Libraries were normalized to 5nM and an equal volume was pooled per library. This final library pool was quantitated by qPCR using the KAPA Library Quantification kit (Roche Diagnostics) and diluted to 2nM for sequencing in 10mM TrisHCl, pH 8.5. Libraries were loaded at 12pM with a 20% phiX spike in on the MiSeq v3, 600 cycle kit according to Illumina's guidelines, generating 2x250 reads.

Analysis of SARS-CoV-2 genomic sequence. We implemented a SARS-CoV-2 genome analysis pipeline that started with raw sequence data and generated quality control information, consensus genomes using the Chan Zuckerberg Biohub SARS-CoV-2 Illumina Pipeline (<https://github.com/czbiohub/sc2-illumina-pipeline>). Consensus genome lineage assignments were created using both Nextclade (v.2.9.1) and Pangolin (v.4.1.3) [12,13]. Per run phylogenetic trees were generated using Augur and visualized in Microreact [14,15].

Data Availability. Whole viral genomes were made publicly available at GenBank (**Supplemental Table 1**).

RESULTS

SARS-CoV-2 genome assessment. Illumina sequences obtained using the ARTIC protocol were processed through our customized SARS-CoV-2 genome analysis pipeline (**Fig 1**). This workflow generates consensus genomes and lineage assignments using both Pangolin and Nextclade. Missing data are assessed using customized plots (**Fig 1B**). Missing data plots were used to assess the genomic location of missing data (basecall = N) due to either poor sequence quality or primer dropout. These plots are useful for assessing if a sample failed to amplify (extensive coloring across the plot) or experienced single primer (repeated pattern) or localized (short stretches across samples) dropouts. **Consensus genomes were excluded from downstream analysis if they were more than 1,000 bases shorter than the 29,903bp SARS-CoV-**

2 Wuhan-Hu-1 reference genome. Interactive phylogenetic trees (**Fig 1C**) are also created and visualized on Microreact (<https://microreact.org/>). In total, we performed ARTIC sequencing on 1,540 samples from which 1,240 consensus genomes passed this size threshold. These consensus genomes were included in all subsequent analyses.

Regional versus national VOC prevalence. Consensus genome sequences were classified based on their similarity to known VOC. The proportion of VOC identified biweekly were calculated and compared to national proportions as reported by the Centers for Disease Control and Prevention (CDC) (SARS-CoV-2 Variant Proportions) (**Fig 2**). Regional VOC prevalence exhibited a great deal of parity with national rates. During the earliest phases of the pandemic (Winter through mid-Summer of 2022) genomes with similarity to the Wuhan-Hu-1 reference genome were gradually replaced with strains from the Alpha lineage. During this same period, the Beta lineage of SARS-CoV-2 subtly emerged nationally, but it was not seen in our local genomic surveillance. Delta was first observed nationally during the first 2-weeks of May 2021 but regional detection was slightly delayed until the last 2-weeks. Other than a limited amount of Alpha lineage detection, the Delta lineage had completely taken over both regionally and nationally by mid-July 2021. Omicron was detected both regionally and nationally in early December and completely replaced the Delta variant by the end of January, 2022. Omicron remained the dominant VOC throughout the Spring and Summer of 2022.

Monitoring regional SARS-CoV-2 subvariant lineages. SARS-CoV-2 VOC lineages are composed of a collection of subvariant lineages. In particular, early pandemic spread of the Omicron lineage was characterized by two primary subvariant lineages. Omicron subvariant lineage BA.1 dominated 2021, with subvariant BA.2 emerging during the Winter of 2021/2022 [7]. Our regional genomic surveillance identified similar patterns (**Fig 3**). The only local Omicron subvariants identified between December, 2021 and February 2022 belonged to the BA.1 lineage (BA1, BA1.1, BA.1.1.10, BA.1.1.4 and BA.1.15). Omicron subvariant BA.2 was first detected in February 2022 completely replacing the BA.1 subvariant lineage by the end of April, 2022. BA.2

was dominant throughout the Summer of 2022 with the dominant subvariants belonging to BA.2 and BA.2.12.1. In total 17 BA.2 subvariants were detected throughout this time period. In addition, Omicron subvariant lineage BA.2 and BA.5 began to be detected in April/May 2022 with increasing detection of BA.5.5 through June.

Analysis of regional genomic sequence divergence from known global references.

Sequence divergence from all locally acquired consensus genomes and selected reference genomes were calculated using Nextclade (**Fig. 4**) [13]. All variants were calculated descendants of previously known reference variants. All local consensus genomes clustering within the Alpha or reference-like clades were classified as descendants to B.1.2, B.1.1.7, BA.1.1.519 and several other B.1.X and B.1.1.X sublineages (**Fig. 4A**). All local consensus genomes assigned to the Delta and Omicron VOC were assigned to the 21A or 21J (**Fig. 4B**) and BA.1, BA.2, BA.4 and BA.5 (**Fig. 4C**) subvariant lineages respectively. No local consensus genomes clustered with WHO reference recombinant genomes.

Analysis of deletions and insertions in regional SARS-CoV-2 genomes. We assessed the prevalence and length of deletions and insertions in all local consensus genomes relative to the SARS-CoV-2 Wuhan-Hu-1 reference genome. We identified a large number of genomic deletions of various frequencies and lengths (**Fig 5, Supplemental Table 2**). High-frequency deletions (> 75% frequency) were identified in all lineages of local consensus genomes other than those assigned to the reference strain (Wuhan-Hu-1) lineage where deletion frequency was relatively low (**Fig 5A**). The number of deletions per lineage varied with genomes assigned to the reference strain (n = 17), Alpha (n = 8), Delta (n=47) and Omicron lineages (21K = 15 and 21L = 19) (**Fig 5**). High-frequency (> 75% frequency) deletions occurred most frequently in ORF1ab and the S gene in genomes assigned to the Alpha, Delta and Omicron lineages. Lineage specific high-frequency deletions were identified in the N gene of Alpha and Omicron (21K and 21L), ORF8 of Delta and the 3' UTR of Omicron 21L. The minimum deletion length across all consensus genomes was 1 base with a maximum deletion length of 126 bases in the Delta lineage within

Orf3a (**Fig 5B**). We identified a cluster of deletions within Orf7a of the Delta lineage (**Fig 5B**). These Delta lineage specific Orf7a deletions had an average start at position 27,588 (min = 27,520, max = 27,721) and an average end at position 27,631 (min = 27,577, max = 27,741) yielding deletions ranging from 1 to 124 bases in length with an average length of 43.5 bases. Each Orf7a deletion was only found in one or two out of 490 local consensus SARS-CoV-2 Delta genomes (lineage frequency range = 0.002% - 0.004%) with the largest 124 base deletion only detected once.

Relative to deletions, far fewer insertions were identified in local consensus genomes (**Fig 6, Supplemental Table 2**). There were no insertions found in consensus genomes assigned to the Alpha lineage. Insertions in the S gene were found in consensus genomes assigned to the reference strain and Omicron strains, but were absent for genomes assigned to the Delta lineage. Insertions ranged in length from 1 to 12 bases, with the longest insertion occurring in the S gene of a single Omicron (BA.2) consensus genome.

DISCUSSION

Pandemics are a global phenomenon comprising a network of local, community driven outbreaks. Understanding global spread starts at the local level. Conversely, understanding how local infection dynamics compare to global trends can provide regional information about the emergence and circulation of globally identified strains. In addition, genomic surveillance can provide highly-detailed genetic maps of regionally circulating strains.

Building a local genomic surveillance program requires access to the appropriate infrastructure including a patient-facing diagnostic lab for detection and procurement of samples containing the pathogen of interest, a molecular biology lab, modern sequencing infrastructure and bioinformaticians for data processing and analysis. Availability to these resources will vary from region-to-region, but a universal consideration will be sampling depth and cadence. Ultimately,

sampling will depend on the goals of the genomic surveillance program as well as financial considerations and laboratory throughput. While access to resources and financing varies across regions, it is still useful to place some boundaries for building a successful sampling framework.

Efforts have been made to model the sampling required to detect the prevalence and emergence of SARS-CoV-2 VOC using regional genomic surveys [16]. These models estimate that VOC detection and average prevalence can be sufficiently measured sequencing ~10 samples/1,000,000 people/week. While models are important guides for designing sampling strategies, real world efforts can be useful for model calibration and validation. In the present study we assessed our ability to detect globally circulating SARS-CoV-2 VOC in a metropolitan hospital in St. Louis, MO, USA. We sampled ~5 samples/1,000,000 people per week (n = 1,240 samples), half of the requirements of the predicted model. We detected all significant SARS-CoV-2 VOC (Alpha, Delta and Omicron) and were also able to detect sublineage variation circulating within the community. Our data support that low number consistent sampling is sufficient to detect the prevalence of known and novel VOC within a community.

In addition to determining local VOC prevalence, regional genomic surveillance provides a wealth of detailed genetic information. While low-frequency subvariants may only appear transiently in a region and never achieve global spread, they still contain genomic variation useful for mechanistic studies. For example, a 115 base pair deletion in SARS-CoV-2 ORF7a (27,549–27,644 nt) was identified as part of regional genomic sequencing efforts in Bozeman, Montana, USA [17]. This ORF7a deletion (ORF7a^{Δ115}) was subsequently shown to have an *in vitro* growth defect associated with elevated IFN response suggesting an immunosuppressive role for ORF7a. In our study, we also identified a collection of deletions in the Orf7a gene in strains from the Delta lineage. Deletions in Orf7a were uncommon, but were as large as 124 bases long. These Orf7a

and other subvariant isolates can serve as a natural laboratory bridging real world genetic variation with detailed mechanistic studies.

Taken together, we demonstrate that modest sampling efforts paired with robust infrastructure can provide genomic surveillance of SARS-CoV-2 at a regional level. These efforts can characterize the presence and prevalence of VOC. In addition, regional genomic surveillance generates a repository of unique genetic variation. We believe these results should serve as a guideline for future SARS-CoV-2 genomic surveillance programs.

Funding: This study was funded by NIH and grant (U01 AI151810).

Author Contributions: B.A.P and D.W. conceptualized the study. B.A.P. coordinated all sample collection. L.D., C.F. and L.C. organized sample extraction and sequencing A.J, B.F. and S.A.H. completed all bioinformatic analysis, and figure design and creation. Writing was shared evenly across all authors.

Bibliography

- [1] D. Rusňáková, T. Sedláčková, P. Radvák, M. Böhmer, P. Mišenko, J. Budiš, S. Bokorová, N. Lipková, M. Forgáčová-Jakúbková, T. Sládeček, J. Sitarčík, W. Krampfl, M. Gažiová, A. Kaliňáková, E. Staroňová, E. Tichá, T. Vrábľová, L. Ševčíková, B. Kotvasová, L. Maďarová, S. Feiková, K. Beňová, L. Reizigová, Z. Onderková, D. Ondrušková, D. Loderer, M. Škereňová, Z. Danková, K. Janíková, E. Halašová, E. Nováková, J. Turňa, T. Szemes, Systematic Genomic Surveillance of SARS-CoV-2 Virus on Illumina Sequencing Platforms in the Slovak Republic-One Year Experience, *Viruses*. 14 (2022). <https://doi.org/10.3390/v14112432>.
- [2] N. Zhu, D. Zhang, W. Wang, X. Li, B. Yang, J. Song, X. Zhao, B. Huang, W. Shi, R. Lu, P. Niu, F. Zhan, X. Ma, D. Wang, W. Xu, G. Wu, G.F. Gao, W. Tan, China Novel Coronavirus Investigating and Research Team, A Novel Coronavirus from Patients with Pneumonia in China, 2019, *N. Engl. J. Med.* 382 (2020) 727–733.
- [3] S. Elbe, G. Buckland-Merrett, Data, disease and diplomacy: GISAID's innovative contribution to global health, *Glob Chall.* 1 (2017) 33–46.
- [4] GISAID - hCov19 Variants, (n.d.). <https://gisaid.org/hcov19-variants/> (accessed December 4, 2022).
- [5] M. Chiara, A.M. D'Erchia, C. Gissi, C. Manzari, A. Parisi, N. Resta, F. Zambelli, E. Picardi, G. Pavesi, D.S. Horner, G. Pesole, Next generation sequencing of SARS-CoV-2 genomes: challenges, applications and opportunities, *Brief. Bioinform.* 22 (2021) 616–630.
- [6] G. John, N.S. Sahajpal, A.K. Mondal, S. Ananth, C. Williams, A. Chaubey, A.M. Rojiani, R. Kolhe, Next-Generation Sequencing (NGS) in COVID-19: A Tool for SARS-CoV-2 Diagnosis, Monitoring New Strains and Phylodynamic Modeling in Molecular Epidemiology, *Curr. Issues Mol. Biol.* 43 (2021) 845–867.
- [7] Tracking SARS-CoV-2 variants, (n.d.). <https://www.who.int/activities/tracking-SARS-CoV-2-variants/tracking-SARS-CoV-2-variants> (accessed December 4, 2022).
- [8] Z. Chen, A.S. Azman, X. Chen, J. Zou, Y. Tian, R. Sun, X. Xu, Y. Wu, W. Lu, S. Ge, Z. Zhao, J. Yang, D.T. Leung, D.B. Domman, H. Yu, Global landscape of SARS-CoV-2 genomic surveillance and data sharing, *Nat. Genet.* 54 (2022) 499–507.
- [9] J.R. Tyson, P. James, D. Stoddart, N. Sparks, A. Wickenhagen, G. Hall, J.H. Choi, H. Lapointe, K. Kamelian, A.D. Smith, N. Prystajacky, I. Goodfellow, S.J. Wilson, R. Harrigan, T.P. Snutch, N.J. Loman, J. Quick, Improvements to the ARTIC multiplex PCR method for SARS-CoV-2 genome sequencing using nanopore, *bioRxiv.* (2020). <https://doi.org/10.1101/2020.09.04.283077>.
- [10] A.W. Lambisia, K.S. Mohammed, T.O. Makori, L. Ndwiga, M.W. Mburu, J.M. Morobe, E.O. Moraa, J. Musyoki, N. Murunga, J.N. Mwangi, D.J. Nokes, C.N. Agoti, L.I. Ochola-Oyier, G. Githinji, Optimization of the SARS-CoV-2 ARTIC Network V4 Primers and Whole Genome Sequencing Protocol, *Front. Med.* 9 (2022) 836728.
- [11] N.D. Grubaugh, K. Gangavarapu, J. Quick, N.L. Matteson, J.G. De Jesus, B.J. Main, A.L. Tan, L.M. Paul, D.E. Brackney, S. Grewal, N. Gurfield, K.K.A. Van Rompay, S. Isern, S.F. Michael, L.L. Coffey, N.J. Loman, K.G. Andersen, An amplicon-based sequencing framework for accurately measuring intrahost virus diversity using PrimalSeq and iVar, *Genome Biol.* 20 (2019) 8.
- [12] Á. O'Toole, E. Scher, A. Underwood, B. Jackson, V. Hill, J.T. McCrone, R. Colquhoun, C. Ruis, K. Abu-Dahab, B. Taylor, C. Yeats, L. du Plessis, D. Maloney, N. Medd, S.W. Attwood, D.M. Aanensen, E.C. Holmes, O.G. Pybus, A. Rambaut, Assignment of epidemiological lineages in an emerging pandemic using the pangolin tool, *Virus Evol.* 7 (2021) veab064.
- [13] I. Aksamentov, C. Roemer, E. Hodcroft, R. Neher, Nextclade: clade assignment, mutation

- calling and quality control for viral genomes, *J. Open Source Softw.* 6 (2021) 3773.
- [14] S. Argimón, K. Abudahab, R.J.E. Goater, A. Fedosejev, J. Bhai, C. Glasner, E.J. Feil, M.T.G. Holden, C.A. Yeats, H. Grundmann, B.G. Spratt, D.M. Aanensen, Microreact: visualizing and sharing data for genomic epidemiology and phylogeography, *Microb Genom.* 2 (2016) e000093.
- [15] J. Huddleston, J. Hadfield, T.R. Sibley, J. Lee, K. Fay, M. Ilcisin, E. Harkins, T. Bedford, R.A. Neher, E.B. Hodcroft, Augur: a bioinformatics toolkit for phylogenetic analyses of human pathogens, *J Open Source Softw.* 6 (2021). <https://doi.org/10.21105/joss.02906>.
- [16] A.X. Han, A. Toporowski, J.A. Sacks, M. Perkins, S. Briand, M. van Kerkhove, E. Hannay, S. Carmona, B. Rodriguez, E. Parker, B.E. Nichols, C.A. Russell, Low testing rates limit the ability of genomic surveillance programs to monitor SARS-CoV-2 variants: a mathematical modelling study, *medRxiv.* (2022). <https://doi.org/10.1101/2022.05.20.22275319>.
- [17] A. Nemudryi, A. Nemudraia, T. Wiegand, J. Nichols, D.T. Snyder, J.F. Hedges, C. Cicha, H. Lee, K.K. Vanderwood, D. Bimczok, M.A. Jutila, B. Wiedenheft, SARS-CoV-2 genomic surveillance identifies naturally occurring truncation of ORF7a that limits immune suppression, *Cell Rep.* 35 (2021) 109197.

Figure Legends:

Figure 1. A) SARS-CoV-2 genome analysis pipeline. **B)** “Dropout Assessment” Location of missing data (N’s) within a selected subset of SARS-CoV-2 consensus genomes. Colors indicate missing data in specific Pangolin lineage assignments. **C)** Single run phylogenetic tree of SARS-CoV-2 consensus genomes visualized using Microreact.

Figure 2. A) National VOC prevalence as reported by the Centers for Disease Control and Prevention. **B)** Regional VOC prevalence as detected in the current study.

Figure 3. Detection of local SARS-CoV-2 Omicron subvariant lineages between December 2021 and June 2022.

Figure 4. A) Phylogenetic tree representing regional sequence divergence in relation to global reference VOC reference genomes. Zoomed in representation of local sequence variants in relation to global reference genomes for **B)** Delta and **C)** Omicron lineages.

Figure 5. A) Lineage specific deletion frequency and **B)** deletion length in local consensus genomes.

Figure 6. A) Lineage specific insertion frequency and **B)** length in local consensus genomes.

B)

C)

This work evaluated the genomic sequencing effort necessary in a regional context to track and detect new SARS-CoV-2 variants. The question is relevant because sufficient sequencing effort would allow timely identification of new variants with potential epidemiological implications. It would also allow informed political decision-making that limits waves of infection. The hypothesis driving the current study was that the sequencing of approximately 5 samples per 100,000 people per week is sufficient to track the regional prevalence and emergence of variants of concern. This hypothesis is related to the work of Han et al. (2022) [1]; these authors estimated the percentage of positive tests that would be necessary to sequence to detect new variants in the material context of low- and middle-income countries. These authors suggest that in a scenario of reduced spatial-temporal bias in sampling, sequencing 5–10% of positive cases would be a good balance between detection time and resource investment.

The authors of the current work made a significant sequencing effort and the emergence of analyses that empirically evaluate epidemiological models is plausible and necessary. However, according to the methodological approach implemented in the current work, the authors did not design a rigorous analysis that would allow formally evaluating what percentage of positive cases per 1,000,000 people per week would need to be sequenced to follow the prevalence and identify new VOCs in a context regional. On the contrary, the authors assume that genomic sequencing of 5 samples/1,000,000 people/week is sufficient and base several of the most important statements of the results on the similarity of the abundance profile of the variants estimated at the regional level with the national profile. For example, the authors make the following statement:

« Here we report the results of a regional SARS-CoV-2 genomic surveillance program run at a metropolitan hospital in St. Louis, MO, USA at a sampling of ~5 samples/1,000,000 people/week (n = 1,240). Our findings provide evidence that this sampling rate is sufficient to detect the prevalence of known and novel VOC within a community.

However, the methodological approach of the current study does not support these statements. Among the main limitations is that this study does not have a ground truth that allows determining what sequencing effort is sufficient. Although the prevalence profile of VOCs at the national level is a good reference, there are differences between the prevalence and the appearance of new variants between the regional level and the national level. The authors could have defined different thresholds of sequencing efforts expressed as a percentage of the number of positive cases and determined from which of these thresholds it is possible to detect new variants or at which of these thresholds the proportion of dominant variants stabilizes. In the analytical framework of the current work, it is not possible to affirm that ~5 samples/1,000,000 people/week is sufficient on a regional scale for the detection of new variants. In the best of cases, it can be said that with this sampling effort, the regional prevalence profile is similar to the national profile.

There are some other suggestions for this work:

- I think that an important piece of data that is not presented in the publication is the percentage of positive cases sequenced. We do not know if the sequencing rate (5 samples/1,000,000 people/week) in the specific context of the current study is at the threshold of 5 to 10% of positive cases, as proposed by Han et al. (2022). Furthermore, presenting the sequencing rate is more robust from an epidemiological and statistical point of view.
- The publication does not discuss some information that may be important to understand the context of this study. For example, the population of the city in which the study was carried out and the importance of the hospital at a regional level. The time between sample collection and genome submission is also not mentioned.
- It is necessary to describe the methodology used to identify indels in genomic sequences.

I think the authors could use the data in another way to provide more significant analytical support for their claims.

1. <https://doi.org/10.1101/2022.05.20.22275319>

Tracking the prevalence and emergence of SARS CoV2 variants of concern using a regional genomic surveillance program

Response to reviewers

Reviewer #1 (Comments for the Author):

Thank you for the opportunity to review this interesting study. The manuscript is well-written, and contributes valuable insights to SARS-CoV-2 genomic surveillance. The authors have presented a comprehensive study on SARS-CoV-2 genomic surveillance at a metropolitan hospital in the USA from February 2021 to June 2022. Their findings highlight the significance of consistent daily sampling in tracking regional prevalence and the emergence of variants of concern (VOC). However, the discussion section needs some improvements.

Strengths:

- The study design is comprehensive, covering a substantial timeframe and utilizing robust methods for sample collection, sequencing, and analysis.
- The methods section is well-detailed, providing clarity on ethical considerations, sample collection, sequencing protocols, and data analysis pipelines.
- The authors conducted a detailed analysis of the SARS-CoV-2 genomic sequences, including classification of VOCs, comparison with national prevalence, identification of sub_variant lineages, and analysis of genetic variations.
- The manuscript emphasizes the importance of regional genomic surveillance for public health preparedness and response. The findings contribute to understanding the prevalence and dynamics of SARS-CoV-2 variants at the local level.

Areas for Improvement:

1. The manuscript could benefit from discussion on the challenges and potential strategies for implementing genomic surveillance in low-resource settings. This would add a broader perspective to the study's implications.

We wholeheartedly agree with the reviewer. This was briefly suggested in the current 2nd paragraph of the discussion with the following sentences:

“Building a local genomic surveillance program requires access to the appropriate infrastructure including a patient-facing diagnostic lab for detection and procurement of samples containing the pathogen of interest, a molecular biology lab, modern sequencing infrastructure and bioinformaticians for data processing and analysis. Availability to these resources will vary from region-to-region, but a universal consideration will be sampling depth and cadence. Ultimately, sampling will depend on the goals of the genomic surveillance program as well as financial considerations and laboratory throughput. While access to resources and financing varies across regions, it is still useful to place some boundaries for building a successful sampling framework.”

However, this only described the challenges. We have added the following statement reflecting on these challenges and how the implications for addressing these challenges at the end of the discussion with the following statements:

“We believe these findings are particularly important when considering setting up surveillance programs in low-resource settings. We demonstrate that with a limited sampling of sequences, we were able to recapitulate sequencing efforts performed at a national level. While this may not prove true for the transmission of all diseases, it does indicate that small, regional efforts are worthwhile and may be more achievable in low-resource settings.

2. While the manuscript focuses on whole genome sequencing, a brief discussion on the complementary role of RT-PCR in surveillance would enhance the completeness of the discussion section.

We agree with the reviewer that non-genomic surveillance methods are incredibly important, particularly in low-resource settings. We discuss how qRT-PCR was used to select samples for full genome analysis. However, this is an important point when considering effective surveillance study design, so we have added the following comments to emphasize this very important and valuable technique:

Non-genomics methods, such as RT-PCR, which can readily detect the presence of SARS-CoV-2, should also be considered during the study design process. This method lacks the ability to resolve base-level sequence variation across the genome. Still these techniques are incredibly efficient in detecting the presence of a virus at considerable cost savings and efficiency.

3. The manuscript could provide more specific recommendations for future SARS-CoV-2 genomic surveillance programs, offering guidance on optimal sampling strategies and adaptation to local resource capacities.

While we agree with the reviewer that this would be useful, we hesitate to provide specific recommendations. Our rationale for excluding specific recommendations is that viral disease surveillance is impacted by innumerable factors (e.g., the type of virus, infectious dose, population-level risk factors, etc.). We maintain that this is a very specific story about the value of moderate surveillance specifically for the spread of SARS-CoV-2 and a reflection on our experiences performing this analysis in a major urban center in the USA.

However, we believe that the earlier reviewer comments about how these findings may be valuable for investigators in low resource settings as well as suggesting investigators consider alternative techniques such as RT-PCR address two important variables (sampling and technology) that would need to be considered in any setting and therefore think that by addressing the reviewers first two queries have partially addressed this third comment.

Few comments are provided as notes in the attached PDF file.

These have been addressed in the resubmitted version of the manuscript.

Reviewer #2 (Comments for the Author):

This work evaluated the genomic sequencing effort necessary in a regional context to track and detect new SARS-CoV-2 variants. The question is relevant because sufficient sequencing effort would allow timely identification of new variants with potential epidemiological implications. It would also allow informed political decision-making that limits waves of infection. The hypothesis driving the current study was that the sequencing of approximately 5 samples per

100,000 people per week is sufficient to track the regional prevalence and emergence of variants of concern. This hypothesis is related to the work of Han et al. (2022) [1]; these authors estimated the percentage of positive tests that would be necessary to sequence to detect new variants in the material context of low- and middle-income countries. These authors suggest that in a scenario of reduced spatial-temporal bias in sampling, sequencing 5–10% of positive cases would be a good balance between detection time and resource investment.

The authors of the current work made a significant sequencing effort and the emergence of analyses that empirically evaluate epidemiological models is plausible and necessary. However, according to the methodological approach implemented in the current work, the authors did not design a rigorous analysis that would allow formally evaluating what percentage of positive cases per 1,000,000 people per week would need to be sequenced to follow the prevalence and identify new VOCs in a context regional. On the contrary, the authors assume that genomic sequencing of 5 samples/1,000,000 people/week is sufficient and base several of the most important statements of the results on the similarity of the abundance profile of the variants estimated at the regional level with the national profile. For example, the authors make the following statement:

« Here we report the results of a regional SARS-CoV-2 genomic surveillance program run at a metropolitan hospital in St. Louis, MO, USA at a sampling of ~5 samples/1,000,000 people/week (n = 1,240). Our findings provide evidence that this sampling rate is sufficient to detect the prevalence of known and novel VOC within a community.

However, the methodological approach of the current study does not support these statements. Among the main limitations is that this study does not have a ground truth that allows determining what sequencing effort is sufficient. Although the prevalence profile of VOCs at the national level is a good reference, there are differences between the prevalence and the appearance of new variants between the regional level and the national level. The authors could have defined different thresholds of sequencing efforts expressed as a percentage of the number of positive cases and determined from which of these thresholds it is possible to detect new variants or at which of these thresholds the proportion of dominant variants stabilizes. In the analytical framework of the current work, it is not possible to affirm that ~5 samples/1,000,000 people/week is sufficient on a regional scale for the detection of new variants. In the best of cases, it can be said that with this sampling effort, the regional prevalence profile is similar to the national profile.

We agree with the overall assessment of the reviewer and have made significant changes to the manuscript to de-emphasize our ability to determine the necessary sequencing effort. Our initial goal was to identify a model for comparison (the Han et al. 2022 manuscript provided this estimate) to assess if we were within a reasonable range to draw any relevant conclusions from our sequencing efforts. However, in agreement with the reviewer we did not sufficiently design a study to systematically affirm that ~5 samples/1e6 people/week is **sufficient** on a regional scale to detect new variants.

Therefore, we have removed all language, suggesting that our data and analysis are sufficient to draw this conclusion. Instead, we have focused on the reviewers' final comment above that "with this sampling effort, the regional prevalence profile is similar to the national profile." We still believe that while this is less powerful than experimentally defining sampling sufficiency, it is still valuable to report. Particularly as an example of how low/moderate sampling may reflect regional or national variant spread, which may be important for laboratories in low-resource

settings. Additional comments emphasizing this have been added to the discussion and are included below:

“We believe these findings are particularly important when considering setting up surveillance programs in low-resource settings. We demonstrate that with a limited sampling of sequences, we were able to recapitulate sequencing efforts performed at a national level. While this may not prove true for the transmission of all diseases, it does indicate that small, regional efforts are worthwhile and may be more achievable in low-resource settings. Non-genomics methods, such as RT-PCR, which can readily detect the presence of SARS-CoV-2, should also be considered during the study design process. This method lacks the ability to resolve base-level sequence variation across the genome. Still these techniques are incredibly efficient in detecting the presence of a virus at considerable cost savings and efficiency.”

There are some other suggestions for this work:

- I think that an important piece of data that is not presented in the publication is the percentage of positive cases sequenced. We do not know if the sequencing rate (5 samples/1,000,000 people/week) in the specific context of the current study is at the threshold of 5 to 10% of positive cases, as proposed by Han et al. (2022). Furthermore, presenting the sequencing rate is more robust from an epidemiological and statistical point of view.

We have added the following text to the Materials and Methods: “From 2/1/21 through 6/30/22, 27,886 positive tests were recorded through various testing methods in the clinical labs. Sequencing selection was based on samples with sufficient viral load as determined by testing on the Roche cobas 6800 instruments (Roche) according to the manufacturer’s instructions. These instruments performed the majority of testing and demonstrated 18,125 positive results during this same time period. The number of sequenced variants, 1,240, represents 6.8% of positive results from the instruments the samples were collected on and 4.4% of all positive tests handled in the clinical lab.”

- The publication does not discuss some information that may be important to understand the context of this study. For example, the population of the city in which the study was carried out and the importance of the hospital at a regional level. The time between sample collection and genome submission is also not mentioned.

We added the following information about the hospital that organized the sampling and the population statistics for the region.

" Barnes-Jewish Hospital is a 1,400-bed nonprofit teaching hospital - the largest in Missouri. It services the St. Louis metropolitan area (population 2.8 million, the 21st largest city in the US as of 2020)."

- It is necessary to describe the methodology used to identify indels in genomic sequences.

The following sentence was added to clarify how polymorphisms, insertions and deletions were identified:

“Polymorphisms, insertions and deletions were determined using the default settings in Minimap2 as implemented in the CZBiohub pipeline.”

I think the authors could use the data in another way to provide more significant analytical support for their claims.

1. <https://doi.org/10.1101/2022.05.20.22275319>

We agree with the reviewer that simulation studies such as the one cited can provide valuable estimates on sample size recommendations. However, the referenced study is entirely simulation, while our study utilizes real-world data. Therefore, we view these as complimentary studies.

Re: Spectrum04225-23R1 (Tracking the prevalence and emergence of SARS CoV2 variants of concern using a regional genomic surveillance program)

Dear Dr. David Wang:

Your manuscript has been accepted, and I am forwarding it to the ASM production staff for publication. Your paper will first be checked to make sure all elements meet the technical requirements. ASM staff will contact you if anything needs to be revised before copyediting and production can begin. Otherwise, you will be notified when your proofs are ready to be viewed.

Data Availability: ASM policy requires that data be available to the public upon online posting of the article. Please add the identifier of the NCBI BioProject for access to the raw reads of this study and please verify all links to sequence records, if present, and make sure that each number retrieves the full record of the data. If a new accession number is not linked or a link is broken, provide production staff with the correct URL for the record. If the accession numbers for new data are not publicly accessible before the expected online posting of the article, publication may be delayed; please contact ASM production staff immediately with the expected release date.

Sincerely,
Holly Ramage
Editor
Microbiology Spectrum

Reviewer #1 (Comments for the Author):

Thank you for opportunity to review the revised manuscript.
The authors addressed my previous comments and made the required revisions.

Reviewer #2 (Comments for the Author):

Please add the identifier of the BioProject and BioSamples that host the raw reads to the document.